# When Does Orthogonal LoRA Help Retrieval? Spectral Preservation, Alignment, and Operating Regimes

## Abstract

Orthogonal variants of LoRA are typically justified as preserving the geometry of the low-rank adaptation subspace in parameter space. For retrieval embedding models, however, whose performance is evaluated directly in embedding space, it remains unclear whether this parameter-space geometry is itself what matters, or whether the decisive factor is the geometry induced in the resulting embeddings. We present a controlled study of 12 LoRA-family methods on implicit concept retrieval and two BEIR passage-retrieval tasks, combining retrieval metrics with weight-space and embedding-space diagnostics. The comparison shows that standard LoRA often collapses the effective rank of its update, but recovering effective rank alone does not reliably recover retrieval quality.

On our primary compact-encoder setting, the strongest retrieval results arise when two conditions are combined: the update remains orthogonal during training and its initial directions are aligned with the pretrained spectral subspace. Motivated by this finding, we introduce GEOLORA-SPECTRAL, a minimal adapter that combines Stiefel-constrained factors, SVD-aligned initialisation, and a learnable diagonal spectral bridge. GEOLORA-SPECTRAL improves over the main LoRA-family baselines in our primary ELSST setting, while the advantage weakens or disappears on less geometry-sensitive tasks and on the larger backbones we study. We clarifies when orthogonality helps retrieval, provides a controlled instantiation of the identified ingredients, and offers a diagnostic toolkit for future embedding-centric PEFT work.

## 1 Introduction

LoRA-style parameter-efficient fine-tuning is now a standard way to fine-tune language-model-based embedding models. Within this line of work, a growing literature has proposed *orthogonal* variants of LoRA: some initialise the low-rank factors from SVD (singular value decomposition) or QR decompositions, some add soft orthogonality penalties, and others enforce hard manifold constraints throughout training. These methods are typically motivated as preserving information, stabilising optimisation, or protecting subspace structure, but the supporting evidence comes predominantly from generation tasks.

For retrieval, the central question is more basic: *when does orthogonality help, and what exactly does it change?* Retrieval quality is computed from similarities in the embedding space, so changes to the encoder's geometry are directly exposed at evaluation time. This makes retrieval a useful setting for testing whether orthogonal constraints preserve useful structure or merely preserve rank.

A natural hypothesis is that orthogonal LoRA helps because it preserves the effective rank of the update matrix. Standard LoRA parameterises the update as $\Delta \mathbf{W} = \mathbf{BA}$, and unconstrained optimisation allows $\mathbf{A}$ and $\mathbf{B}$ to co-adapt so that the *effective* rank of $\Delta \mathbf{W}$ can become much lower than the nominal rank $r$. If rank collapse is the main failure mode, then any method that prevents it should improve retrieval. This hypothesis is plausible, but it has not been tested systematically across orthogonalisation strategies under a shared retrieval protocol.

We compare 12 LoRA-family methods, organised into five categories according to how strictly they maintain orthogonality during training: unconstrained, initialisation-only, soft-constrained, hard-constrained, and

structural counterfactuals. We pair retrieval metrics with geometric diagnostics, including effective rank, orthogonality error, alignment, uniformity, and isotropy. Our primary testbed is ELSST implicit concept retrieval (UK Data Service, 2020; NLPCC 2026 shared task 7, 2026), a zero-overlap benchmark with a fixed, semantically dense concept pool. Because each query has only 1–5 positives among 3,433 candidate concepts, we expect small geometric changes to affect ranking more directly than on passage-retrieval benchmarks, where lexical overlap can partly mask them. We then use two BEIR passage-retrieval tasks as scope tests to assess whether the same pattern persists when retrieval is less geometry-sensitive.

The comparison reveals that standard LoRA often collapses the effective rank of its update, but retrieval quality does not vary monotonically with effective rank. On our primary compact encoder, the strongest retrieval results arise when hard orthogonality is combined with spectral alignment to the pretrained weight matrix. This advantage is not universal: it is clearest at larger $r/d$ and on geometry-sensitive retrieval, and it weakens or disappears on the other backbones and tasks we study.

Motivated by this observation, we introduce GEOLORA-SPECTRAL, a minimal parameterisation that combines hard orthogonality, SVD-aligned initialisation, and a learnable spectral bridge. The paper's contribution is insight-first rather than benchmark-first: GEOLORA-SPECTRAL is designed as a controlled instantiation of the mechanism suggested by the comparative study, not as a universal replacement for LoRA.

**Contributions.** Our contributions are threefold: **(1) Mechanism-level comparison:** We provide a unified evaluation of 12 LoRA-family methods that separates three often-conflated factors: effective-rank preservation, orthogonality during training, and spectral alignment with the pretrained weight. **(2) Clarifying the role of rank:** We show that standard LoRA often collapses effective rank on retrieval models, but recovering rank alone does not determine retrieval quality. Among high-rank methods, performance still varies materially, indicating that additional geometric structure matters. **(3) Minimal controlled instantiation:** We introduce GEOLORA-SPECTRAL, a minimal adapter that combines Stiefel-constrained factors, SVD-aligned initialisation, and a learnable diagonal bridge, and we characterise the regime in which this design is most promising in our experiments.

## 2 Related Work

### 2.1 LoRA for Retrieval and Representation Models

Dense retrieval systems depend heavily on the geometry of the learned embedding space. Early dual-encoder methods such as DPR (Karpukhin et al., 2020) and ANCE (Xiong et al., 2021) established that contrastive training in a shared embedding space can match or surpass sparse retrieval, while subsequent work has shown that hard-negative mining, loss design, and pooling choices are critical for shaping that geometry (Gao et al., 2021; Qu et al., 2021). More recently, LLM-based dense retrievers have pushed the frontier further (Ma et al., 2024), and standardised benchmarks such as MTEB (Muennighoff et al., 2023) have enabled systematic cross-task evaluation of embedding quality.

LoRA-style adaptation is now widely used when those models must be specialised to a new domain, but most work treats the adapter as a lightweight optimisation device rather than as an object whose own geometry may matter. Biderman et al. (2024) provide a large-scale comparison showing that LoRA learns lower-rank perturbations than full fine-tuning and trades off learning capacity for reduced forgetting—but their analysis targets generation, not embedding geometry. Our paper focuses precisely on that missing layer of analysis: how the internal structure of the low-rank update shapes the embedding space under a retrieval objective.

### 2.2 Orthogonal LoRA Variants

A broad family of orthogonal LoRA methods has emerged. PiSSA (Meng et al., 2024) and OLoRA (Büyükakyuz, 2024) impose structure mainly through initialisation. SORSA (Cao & Song, 2024) uses a soft regulariser. Householder-reflection adaptation (Yuan et al., 2024) parameterises the update through products of reflections, guaranteeing orthogonality by structure rather than by constraint. StelLA (Li et al.,

2025), Riemannian LoRA (Bogachev et al., 2025), OPLoRA (Xiong & Xie, 2026), and orthogonal high-rank adaptation (Zhang et al., 2025a) move closer to hard manifold-constrained optimisation, each with a different balance between expressivity and training-time cost. Park et al. (2025) further show that optimising the $B$ matrix on the Stiefel manifold with a dedicated Riemannian optimiser achieves near-perfect orthogonality and full effective rank, yielding consistent gains on generation benchmarks with both LoRA and DoRA. These papers differ in their mathematical machinery and target tasks, but they share an implicit premise: preserving or encouraging orthogonality should improve the behaviour of low-rank adaptation.

What has been missing is a retrieval-side comparison that separates three questions that are often conflated: does a method preserve effective rank, does it maintain orthogonality during training, and does it align the update with the pretrained spectral structure? Our paper is organised around that decomposition.

### 2.3 Geometry of Learned Representations

Several strands of prior work motivate our diagnostic choices. Prior studies of contextual embeddings have shown that representations are often anisotropic (Ethayarajh, 2019). Work on contrastive learning has studied dimensional collapse and the alignment–uniformity trade-off (Jing et al., 2022; Wang & Isola, 2020); Gao et al. (2021) further demonstrate, both theoretically and empirically, that contrastive objectives regularise the anisotropic pretrained embedding space toward greater uniformity, a property central to our own diagnostics. More recent analyses argue that singular directions, norms, and subspace alignment can affect how representations interact after fine-tuning or model merging (Zhang et al., 2025b; Tang et al., 2024). In particular, Zhang et al. (2025b) show that LoRA fine-tuning introduces *intruder dimensions* (high-ranking singular vectors dissimilar to those in the pretrained weight matrix) while full fine-tuning does not, and that these intruder dimensions drive catastrophic forgetting. This spectral perspective motivates our investigation of whether SVD-aligned initialisation can mitigate such spectral disruption in the retrieval setting. Our paper connects these threads to parameter-efficient adaptation. Rather than studying a single model or a single geometric statistic, we compare a family of LoRA methods under one fixed retrieval protocol and ask which geometric properties actually track retrieval quality.

**Scope.** The comparison is intentionally limited to the LoRA family because the paper's central variable (the orthogonality and spectral alignment of a low-rank weight update) does not have a direct analogue in activation-side PEFT methods such as adapters (Houlsby et al., 2019) or prefix tuning (Li & Liang, 2021). We leave a cross-family comparison (e.g., LoRA-family vs. adapter-family) in retrieval to future work.

## 3 Experimental Setup

We fix a retrieval pipeline and vary only the LoRA-family method. This section describes the common setup used for both the comparative analysis (Section 5) and the downstream evaluation (Section 6).

### 3.1 Models

We evaluate three embedding backbones spanning architecture type, scale, and embedding dimensionality:

- **multilingual-e5-small** (Wang et al., 2024): 118M parameters, bidirectional encoder, 384-dimensional embeddings ($r/d = 2.08\%$ at rank $r = 8$).

- **bge-base-en-v1.5** (Xiao et al., 2024): 109M parameters, bidirectional encoder, 768-dimensional embeddings ($r/d = 1.04\%$).

- **Qwen3-Embedding-0.6B** (Team, 2025): 600M parameters, decoder-based embedding model with 1024-dimensional embeddings ($r/d = 0.78\%$).

This choice lets us examine whether the behaviour of orthogonal LoRA changes with model scale, architecture, and the relative size of the rank budget.

## 3.2 Datasets

**ELSST Track 1: Implicit Concept Retrieval** (NLPCC 2026 shared task 7, 2026) is our primary testbed. Each query is a document; the retrieval target is a subset of 3,433 expert-authored concepts ranked by cosine similarity. Train and test concepts have zero overlap, each query has 1–5 positives in a semantically dense pool, and we therefore expect small geometric changes to affect ranking more strongly than on passage-retrieval benchmarks, where lexical overlap can partially mask them. **SciFact** (Wadden et al., 2020) and **NFCorpus** (Boteva et al., 2016) from BEIR (Thakur et al., 2021) serve as scope tests for standard binary passage retrieval.

## 3.3 Baselines

We compare 12 LoRA-family methods spanning five tiers of an *orthogonalisation continuum*, summarised in Table 1. The tiers range from UNCONSTRAINED methods that impose no restriction on the update trajectory, through INITIALISATION-ONLY and SOFT-CONSTRAINED methods that encourage orthogonality at the start or via a penalty, to HARD-CONSTRAINED methods that enforce strict orthogonality throughout training. Two STRUCTURAL counterfactuals that replace the matrix product with Hadamard or Kronecker operations complete the comparison. All methods share the training recipe described in Section 3.4; performance differences therefore reflect the parameterisation itself, not optimiser engineering.

Table 1: Overview of the 12 LoRA-family methods compared in this paper, organised by the *orthogonalisation continuum*. The three binary columns correspond to the analytic axes: whether the method preserves effective rank (**Rank**), maintains orthogonality throughout training (**Ortho.**), and aligns the update with the pretrained spectral structure (**Spectral**). ✓ = yes; ✗ = no; $\sim$ = partial; N/A = not directly comparable. $\odot$: Hadamard product; $\otimes$: Kronecker product.

| Tier | Method | Update parameterisation | Rank | Ortho. | Spectral | Reference |
|---|---|---|---|---|---|---|
| UNCONSTRAINED | LoRA | $\Delta W = BA$; Gaussian / zero init | ✗ | ✗ | ✗ | Hu et al. (2022) |
| | DoRA | Magnitude–direction decomp.; LoRA on direction | ✗ | ✗ | ✗ | Liu et al. (2024) |
| | AdaLoRA | $P\Lambda Q$; importance-aware rank pruning | ✗ | ✗ | $\sim$ | Zhang et al. (2023) |
| INIT-ONLY | PiSSA | $\Delta W = BA$; $A, B$ from top-$r$ SVD of $W$ | ✓ | ✗ | ✓ | Meng et al. (2024) |
| | OLoRA | $\Delta W = BA$; $A, B$ from QR of $W$ | ✓ | ✗ | ✗ | Büyükakyuz (2024) |
| SOFT | SORSA | SVD init + $\lambda\|AA^\top - I_r\|_F$ penalty | ✓ | $\sim$ | ✓ | Cao & Song (2024) |
| HARD | Stiefel-LoRA | $USV^\top$; $U, V \in$ St; Riemannian optim. | ✓ | ✓ | ✗ | Li et al. (2025) |
| | OPLoRA | Orthogonal-projection constraint per step | ✓ | ✓ | ✗ | Xiong & Xie (2026) |
| | GeoLoRA-Ortho | $B \operatorname{diag}(\boldsymbol{\sigma}) A$; random Stiefel init | ✓ | ✓ | ✗ | Ours |
| | GeoLoRA-Spectral | $B \operatorname{diag}(\boldsymbol{\sigma}) A$; SVD-aligned init | ✓ | ✓ | ✓ | Ours |
| STRUCTURAL | LoHA | $\Delta W = (A_1 B_1) \odot (A_2 B_2)$ | N/A | ✗ | ✗ | Hyeon-Woo et al. (2021) |
| | LoKr | $\Delta W = A \otimes B$ | N/A | ✗ | ✗ | Bai et al. (2026) |

**Comparison scope.** All methods share the training recipe in Section 3.4; we do not reproduce bespoke optimisers or schedules from individual method papers. Performance differences therefore reflect the parametrisation under a shared protocol rather than optimiser engineering.[1]

## 3.4 Training Protocol

All methods share the same concept-retrieval fine-tuning recipe. Each training example pairs a document with the description of one of its gold concepts as the positive and uses 8 negatives (4 dataset-provided hard

---

[1] Two implementation notes follow from this design choice. (i) Our SORSA baseline retains the soft orthogonality regulariser (verified to remain numerically active throughout training) but uses AdamW rather than SORSA's SVD-aware optimiser; the contribution we test is therefore "soft penalty under the shared optimiser." (ii) Our Stiefel-LoRA baseline enforces the hard Stiefel constraint via `torch.nn.utils.parametrize` with AdamW, which approximates the Riemannian optimisation used by Li et al. (2025). We therefore interpret any underperformance of these two baselines as evidence about the parametrisation under our shared recipe rather than about the original papers, and we do not claim that our GeoLoRA variants outperform StelLA or SORSA under their native training setups.

negatives and 4 random negatives) under an InfoNCE loss (van den Oord et al., 2018) with temperature $\tau = 0.05$. We train for 5 epochs with batch size 8, learning rate $10^{-4}$, cosine decay, rank $r = 8$, and scaling factor $\alpha = 8$. Adapters target the query, key, and value projection layers in every transformer block. Unless stated otherwise, each experiment is repeated with three seeds (42, 43, 44), and we report the mean and standard deviation.

**Frozen-model calibration.** To verify that ELSST is not trivially solved by the pretrained representation, we also evaluate the unfine-tuned backbones. Their Recall@5 values are substantially below the fine-tuned models (e5-small: 0.211, bge-base: 0.272, Qwen3: 0.278), confirming that the benchmark leaves meaningful room for adaptation.

## 3.5 Metrics

**Retrieval metrics.** We report Recall@k ($k \in \{5, 10, 20\}$), MAP, MRR, and NDCG@5. Recall@5 is the primary metric on ELSST because each query can have multiple relevant concepts, whereas MRR is primary for the BEIR scope tests because those tasks are standard binary passage retrieval. For a query $q$ in the evaluation set $Q$, let $\mathcal{R}_q$ denote its set of gold-relevant documents, and let $\text{Top}_k(q)$ denote the set of the top-$k$ candidates returned by the model. We define

$$\text{Recall@}k = \frac{1}{|Q|} \sum_{q \in Q} \frac{|\mathcal{R}_q \cap \text{Top}_k(q)|}{|\mathcal{R}_q|},$$

where $|\cdot|$ denotes set cardinality.

**Geometry diagnostics.** We track five metrics across two levels.

**Weight-space** metrics evaluate the adapter update $\Delta\mathbf{W} \in \mathbb{R}^{d_{\text{out}} \times d_{\text{in}}}$. For BA-style methods, $\Delta\mathbf{W} = \mathbf{BA}$ where $\mathbf{B} \in \mathbb{R}^{d_{\text{out}} \times r}$ and $\mathbf{A} \in \mathbb{R}^{r \times d_{\text{in}}}$ are learned low-rank factors; GEOLORA-SPECTRAL uses the diagonally scaled variant $\Delta\mathbf{W} = \mathbf{B} \text{diag}(\boldsymbol{\sigma})\mathbf{A}$. Here $d_{\text{in}}$ and $d_{\text{out}}$ are the input and output dimensions, and $r$ is the nominal rank:

(i) **Effective rank** (Roy & Vetterli, 2007): $\text{EffRank}(\Delta\mathbf{W}) = \exp(-\sum_{i=1}^{r} p_i \log p_i)$, where $p_i = \sigma_i^2 / \sum_{j=1}^{r} \sigma_j^2$ and $\sigma_i$ are the singular values of $\Delta\mathbf{W}$. Values near the nominal rank $r$ indicate full rank utilisation; values $\ll r$ imply rank collapse. (We apply the identical formula to the eigenvalues of the embedding covariance matrix $\mathbf{C}$).

(ii) **Orthogonality error**: $\text{OrthoErr}(\mathbf{A}) = \|\mathbf{AA}^\top - \mathbf{I}_r\|_F$, where $\mathbf{I}_r$ is the $r \times r$ identity matrix and $\|\cdot\|_F$ denotes the Frobenius norm. This measures the deviation from row-orthonormality. For hard-constrained methods it should be numerically close to zero.

**Embedding-space** metrics evaluate the $\ell_2$-normalised encoder output $f(\mathbf{x})$ over $n$ test samples. Let $p_{\text{pos}}$ denote the distribution of positive query-document pairs $(\mathbf{x}, \mathbf{x}^+)$ and $p_{\text{data}}$ denote the marginal data distribution over all valid items:

(iii) **Alignment** ($\mathcal{A}$) and (iv) **Uniformity** ($\mathcal{U}$) (Wang & Isola, 2020):

$$\mathcal{A} = \mathop{\mathbb{E}}_{(\mathbf{x}, \mathbf{x}^+) \sim p_{\text{pos}}} \left[ \|f(\mathbf{x}) - f(\mathbf{x}^+)\|^2 \right], \quad \mathcal{U} = \log \mathop{\mathbb{E}}_{\mathbf{x}, \mathbf{y} \sim p_{\text{data}}} \left[ e^{-2\|f(\mathbf{x}) - f(\mathbf{y})\|^2} \right].$$

Lower $\mathcal{A}$ indicates tighter clustering of positive pairs, while a more negative $\mathcal{U}$ reflects a more uniform spread of independent samples $\mathbf{x}$ and $\mathbf{y}$ on the hypersphere.

(v) **Isotropy**: Defined as the mean pairwise cosine similarity, $\text{Iso} = \frac{2}{n(n-1)} \sum_{i<j} f(\mathbf{x}_i)^\top f(\mathbf{x}_j)$ (Ethayarajh, 2019), where $\mathbf{x}_i$ and $\mathbf{x}_j$ represent the $i$-th and $j$-th items in the test set. Values near zero denote isotropic spread; values $\gg 0$ indicate anisotropic collapse.

**Statistical testing.** Significance is assessed via paired bootstrap confidence intervals over per-query deltas (10,000 resamples).

# 4 A Minimal Controlled Instantiation: GeoLoRA-Spectral

The questions posed in Section 1 (whether orthogonality helps retrieval, and what it actually changes) require a parameterisation that separates rank preservation, orthogonality maintenance, and spectral alignment as cleanly as possible. Standard LoRA conflates all three: its unconstrained $\Delta\mathbf{W} = \mathbf{BA}$ factorisation allows rank to collapse, orthogonality to drift, and initialisation to be arbitrary, making attribution difficult.

We therefore design GeoLoRA-Spectral as a *minimal controlled* architecture in which each ingredient corresponds to an isolated design choice: hard orthogonality is enforced by constraining $\mathbf{A}$ and $\mathbf{B}$ to the Stiefel manifold, spectral alignment is controlled through initialisation (SVD-aligned vs. random), and per-direction magnitude is governed by a learnable diagonal $\boldsymbol{\sigma}$ that decouples scaling from subspace selection. The ablation variant GeoLoRA-Ortho replaces SVD-aligned initialisation with random Stiefel initialisation, yielding a controlled pair that differs in exactly one variable.

## 4.1 Architecture

For a target layer with pretrained weight $\mathbf{W} \in \mathbb{R}^{d_{\text{out}} \times d_{\text{in}}}$, GeoLoRA-Spectral uses

$$\mathbf{W}' = \mathbf{W} + \frac{\alpha}{r} \mathbf{B} \, \text{diag}(\boldsymbol{\sigma}) \, \mathbf{A}, \tag{1}$$

where $\mathbf{A} \in \mathbb{R}^{r \times d_{\text{in}}}$ satisfies $\mathbf{AA}^\top = \mathbf{I}_r$, $\mathbf{B} \in \mathbb{R}^{d_{\text{out}} \times r}$ satisfies $\mathbf{B}^\top \mathbf{B} = \mathbf{I}_r$, and $\boldsymbol{\sigma} \in \mathbb{R}^r$ is a learnable vector of per-direction magnitudes. Equivalently, $\mathbf{A}^\top \in \text{St}(d_{\text{in}}, r)$ and $\mathbf{B} \in \text{St}(d_{\text{out}}, r)$, where $\text{St}(n, r)$ denotes the Stiefel manifold of $n \times r$ matrices with orthonormal columns. The operator $\text{diag}(\cdot) : \mathbb{R}^r \to \mathbb{R}^{r \times r}$ maps $\boldsymbol{\sigma}$ to a diagonal matrix. Relative to standard LoRA, this parameterisation makes the separation between *subspace directions* and *per-direction magnitudes* explicit; with $\alpha/r = 1$ as throughout the paper, $\boldsymbol{\sigma}$ is the sole source of per-direction scaling.

The forward pass avoids materialising the dense update: $\mathbf{z} = \mathbf{Ax}$, $\mathbf{z}' = \boldsymbol{\sigma} \odot \mathbf{z}$, $\Delta\mathbf{y} = \mathbf{Bz}'$. Inference cost is $\mathcal{O}(r(d_{\text{in}} + d_{\text{out}}))$, identical to LoRA after merging. In our implementation, training adds roughly 5% wall-clock overhead at $r=8$ from the orthogonal reparametrisation. Hard orthogonality is implemented via PyTorch's built-in Stiefel parametrisation.

## 4.2 SVD-Aligned Initialisation

GeoLoRA-Spectral initialises its factors from the truncated SVD of the pretrained weight. Algorithm 1 summarises the procedure.

---

**Algorithm 1** SVD-aligned initialisation for GeoLoRA-Spectral. In line 4, diag($\mathbf{S}$) extracts the vector of diagonal entries of the diagonal matrix $\mathbf{S}$; the result is the learnable vector $\boldsymbol{\sigma}$ of per-direction magnitudes.

---

**Require:** Pretrained weight $\mathbf{W} \in \mathbb{R}^{d_{\text{out}} \times d_{\text{in}}}$, rank $r$, scale $s$
 1: Compute truncated SVD: $\mathbf{W} \approx \mathbf{USV}^\top$ where $\mathbf{U} \in \mathbb{R}^{d_{\text{out}} \times r}$, $\mathbf{S} \in \mathbb{R}^{r \times r}$, $\mathbf{V} \in \mathbb{R}^{d_{\text{in}} \times r}$
 2: $\mathbf{A} \leftarrow \mathbf{V}^\top$
 3: $\mathbf{B} \leftarrow \mathbf{U}$
 4: $\boldsymbol{\sigma} \leftarrow s \cdot \text{diag}(\mathbf{S})$    (extracts the diagonal of $\mathbf{S}$ as a vector of length $r$)
**Ensure:** $\mathbf{AA}^\top = \mathbf{I}_r$, $\mathbf{B}^\top \mathbf{B} = \mathbf{I}_r$

---

This initialisation places the adapter in the leading singular subspaces of the pretrained weight. We use scale $s = 0.1$; smaller values degrade MRR (Section 6.2).

**Design simplicity.** GeoLoRA-Spectral uses diagonal (rather than full $r \times r$) scaling and requires no custom Riemannian optimiser; the aim is to instantiate, with minimal extra machinery, the two ingredients highlighted by the comparative study.

**Ablation variant.** GeoLoRA-Ortho replaces SVD-aligned initialisation with random Stiefel initialisation, isolating the value of spectral alignment.

## 5 Geometric Analysis: What Orthogonal LoRA Changes in Retrieval

This section pairs retrieval metrics with the geometric diagnostics of Section 3.5 to ask three questions: does orthogonal LoRA preserve effective rank (Section 5.1), does rank preservation alone explain retrieval quality (Section 5.2), and what additional ingredient is needed for peak performance (Section 5.3)? To disentangle the roles of hard orthogonality and spectral alignment, we use the two controlled probes defined in Section 4: GEOLORA-ORTHO (random init) and GEOLORA-SPECTRAL (SVD-aligned init).

### 5.1 Finding 1: Standard LoRA collapses the effective rank of its update

Table 2 reports the effective rank, orthogonality error, and Recall@5 for all 12 methods across three backbones. The pattern is consistent: unconstrained methods collapse to an effective rank of roughly 4–5 despite nominal rank $r = 8$, using only ∼56% of the rank budget on e5-small, ∼52% on bge-base, and ∼61% on Qwen3. Hard-constrained methods maintain both full effective rank ($\geq 7.88$) and near-zero orthogonality error ($< 10^{-6}$) on all three backbones. Initialisation-only and soft-constrained methods retain high effective rank but allow orthogonality to drift substantially during training (OrthoErr $> 6$ on all backbones), occupying a middle ground that does not translate into consistent retrieval gains.

Table 2: Family-wide geometric snapshot across three backbones on ELSST (epoch 5). EffRank, R@5 ,and OrthoErr are reported for seed 42. EffRank: entropy-based effective rank of $\Delta \mathbf{W} = \mathbf{BA}$; OrthoErr $:= \|\mathbf{AA}^\top - \mathbf{I}_r\|_F$. Best R@5 per backbone in **bold**, second-best underlined. — = metric undefined for that parameterisation; * = not applicable (Hadamard/Kronecker updates).

| Tier | Method | e5-small (384d, $r/d$=2.1%) EffRank | OrthoErr | R@5 | bge-base (768d, $r/d$=1.0%) EffRank | OrthoErr | R@5 | Qwen3-0.6B (1024d, $r/d$=0.8%) EffRank | OrthoErr | R@5 |
|---|---|---|---|---|---|---|---|---|---|---|
| *Unc.* | LoRA | 4.50 | 2.22 | 0.3420 | 4.17 | 2.22 | **0.4336** | 4.90 | 2.05 | 0.4490 |
| | DoRA | 4.55 | 2.23 | 0.3412 | 4.19 | 2.22 | 0.4334 | 4.99 | 2.05 | 0.4502 |
| | AdaLoRA | 5.90 | — | 0.3136 | 6.17 | — | 0.4156 | 7.39 | — | 0.4648 |
| *Init* | PiSSA | 7.95 | 10.5 | 0.3532 | 7.93 | 8.79 | 0.4263 | 7.91 | 6.91 | 0.3949 |
| | OLoRA | 7.90 | 13.8 | 0.3448 | 7.86 | 7.73 | 0.4268 | 7.64 | 7.97 | 0.3578 |
| *Soft* | SORSA | 7.95 | 10.4 | 0.3516 | 7.93 | 8.77 | 0.4263 | 7.91 | 6.86 | 0.3965 |
| *Hard* | Stiefel-LoRA | 8.00 | ≈0 | 0.3313 | 8.00 | ≈0 | 0.4256 | 8.00 | ≈0 | 0.4077 |
| | OPLoRA | 8.00 | ≈0 | 0.3413 | 8.00 | ≈0 | 0.4144 | 8.00 | ≈0 | 0.4171 |
| | GEOLORA-O | 7.88 | ≈0 | 0.3405 | 7.89 | ≈0 | 0.4191 | 7.70 | ≈0 | 0.4033 |
| | GEOLORA-S | 7.95 | ≈0 | **0.3567** | 7.92 | ≈0 | 0.4270 | 7.90 | ≈0 | 0.4199 |
| *Str.* | LoHA | * | * | 0.3355 | * | * | 0.4206 | * | * | **0.4689** |
| | LoKr | * | * | 0.2884 | * | * | 0.3967 | * | * | 0.4609 |

### 5.2 Finding 2: Effective rank alone does not explain retrieval quality

If rank collapse were the sole problem, then methods with near-full effective rank should achieve similar retrieval quality. Table 2 does not support that view. On e5-small, methods with EffRank $\geq 7.88$ still span a broad range of Recall@5, from 0.3313 (Stiefel-LoRA) to 0.3567 (GEOLORA-SPECTRAL). Figure 1a visualises the same point: once rank collapse is avoided, the ordering among methods remains non-trivial.

The implication is not that rank is irrelevant. Rather, rank preservation removes one failure mode, but it does not determine which high-rank method will work best. Additional geometric properties are needed to explain why some near-full-rank methods help retrieval and others do not.

### 5.3 Finding 3: The strongest e5-small result combines hard orthogonality with spectral alignment

What distinguishes GeoLoRA-Spectral from the other high-rank methods? A $2 \times 2$ design isolates two ingredients: whether hard orthogonality is maintained throughout training, and whether the adapter is initialised in the pretrained spectral subspace.

**Disentangling orthogonality from spectral alignment.** Four methods populate a natural $2 \times 2$ grid (Table 3), crossing hard orthogonality (off/on) with spectral alignment (off/on). On e5-small, GeoLoRA-Spectral is the strongest of the four cells. Spectral alignment already helps when hard orthogonality is off (LoRA $\rightarrow$ PiSSA), but the highest score is obtained when spectral alignment is combined with hard orthogonality. Hard orthogonality alone does not improve over LoRA, which indicates that preserving the subspace constraint without anchoring the directions to the pretrained spectrum is not enough in this setting.

Table 3: Orthogonality and spectral alignment on e5-small (Recall@5, 3-seed mean). The highest score is obtained when both ingredients are present.

|  | Spectral: off | | Spectral: on | |
| --- | --- | --- | --- | --- |
| **Hard ortho: off** | LoRA | 0.3420 | PiSSA | 0.3532 |
| **Hard ortho: on** | GeoLoRA-Ortho | 0.3405 | GeoLoRA-Spectral | **0.3567** |

**Spectral evidence.** The learned $\boldsymbol{\sigma}$ values provide weight-space corroboration (Figure 1c, d). GeoLoRA-Spectral develops substantially larger and more differentiated per-direction magnitudes than GeoLoRA-Ortho across all layers (panel c), and its $\boldsymbol{\sigma}$ values correlate with the pretrained top-$r$ singular values ($\rho = 0.49$ vs. $\rho = 0.01$ for GeoLoRA-Ortho; panel d). This suggests that SVD-aligned initialisation gives the spectral bridge a meaningful starting point, whereas random Stiefel initialisation leaves the per-direction scaling without a pretrained signal to amplify within our five-epoch budget.

**Alignment paradox.** GeoLoRA-Ortho achieves tighter positive-pair alignment than GeoLoRA-Spectral (Table 4), yet GeoLoRA-Spectral retrieves better. This suggests that local alignment alone is not enough: GeoLoRA-Ortho appears to contract the global embedding distribution more aggressively (higher hubness; Supplementary, Table S6), whereas GeoLoRA-Spectral preserves a more uniform space. We therefore interpret spectral alignment as helping to retain the pretrained model's global organisation rather than merely tightening positive pairs. In the supplementary analysis (§ S5), we also examine update magnitude and distortion and do not find them sufficient to explain the gap on their own.

Table 4: Embedding-space geometry on the ELSST test set. Lower is better for alignment and uniformity; isotropy near zero indicates even directional spread.

| Method | Alignment $\downarrow$ | Uniformity $\downarrow$ | Isotropy |
| --- | --- | --- | --- |
| GeoLoRA-Spectral | 0.995 | $-2.516$ | 0.109 |
| DoRA | 1.065 | $\mathbf{-2.574}$ | 0.124 |
| LoRA | 1.063 | $-2.567$ | 0.123 |
| GeoLoRA-Ortho | **0.842** | $-2.227$ | 0.074 |

**Statistical validation.** Figure 2 provides per-query paired-bootstrap testing on e5-small (10,000 resamples, $N$=5,733 paired observations). GeoLoRA-Spectral is the only method whose 95% CI for $\Delta$MRR lies entirely above zero (panel a), and it achieves the highest per-query win rate (31%, panel b). On bge-base, no method significantly improves over LoRA; on Qwen3, several do (Supplementary, Figure S3).

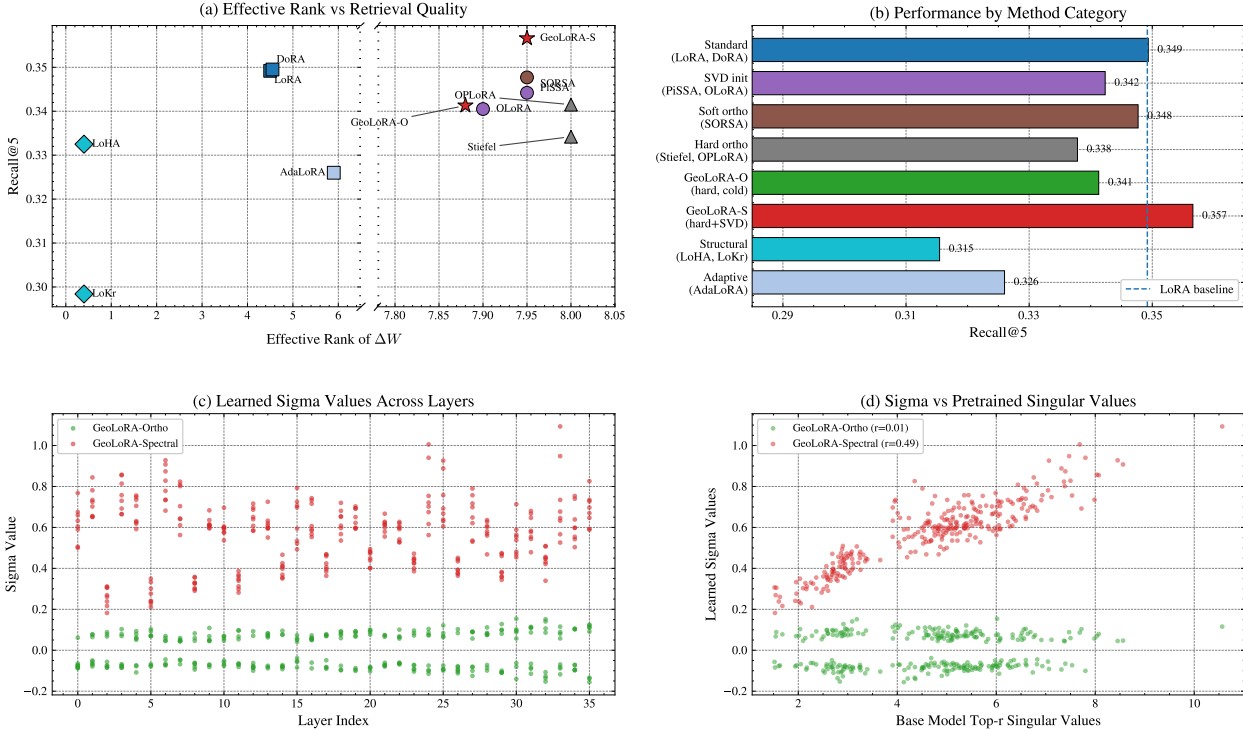

Figure 1: Geometric diagnostics across 12 LoRA variants on e5-small, supporting Findings 2 and 3. **(a)** Effective rank of $\Delta\mathbf{W}$ vs. Recall@5: the relationship is not monotone; high-rank methods still separate in retrieval quality. **(b)** Recall@5 aggregated by method category: the strongest e5-small result comes from the GeoLoRA-Spectral condition, showing that maximal rank alone does not determine performance. **(c)** Learned $\boldsymbol{\sigma}$ values across layers: GEOLoRA-SPECTRAL develops larger and more differentiated per-direction magnitudes than GEOLoRA-ORTHO. **(d)** $\boldsymbol{\sigma}$ vs. pretrained top-$r$ singular values: GEOLoRA-SPECTRAL shows a substantially stronger correlation than GEOLoRA-ORTHO, suggesting that SVD-aligned initialisation lets the spectral bridge exploit pretrained structure.

## 6 Scope, Sensitivity, and Ablations

The geometric analysis of Section 5 shows that the best e5-small result arises when hard orthogonality is combined with spectral alignment. This section asks three follow-up questions: where does that advantage hold (and where does it break), how sensitive is it to rank and initialisation scale, and what is the computational cost?

### 6.1 Operating Regime

**Across backbones.** At fixed $r=8$, the GeoLoRA-Spectral–LoRA gap varies substantially across backbones: it is positive on e5-small, slightly negative on bge-base, and negative on Qwen3 in Table 2. The clearest benefit therefore appears on the compact encoder with the largest $r/d$, and it does not transfer uniformly as model dimension increases. Two non-exclusive explanations are plausible: (i) when $r \ll d$, preserving the full rank of the update may affect too small a fraction of the embedding space to matter; (ii) larger models may distribute task-relevant information across a broader or more redundant spectral basis, making the top-$r$ singular directions a weaker prior for adaptation. The current results are consistent with these hypotheses, but they do not isolate them. The bge-base result is especially instructive: no method in the 12-method panel significantly improves over LoRA on this backbone (Supplementary, Table S4), suggesting, though not proving, that the task can already be solved within a narrower adapted subspace.

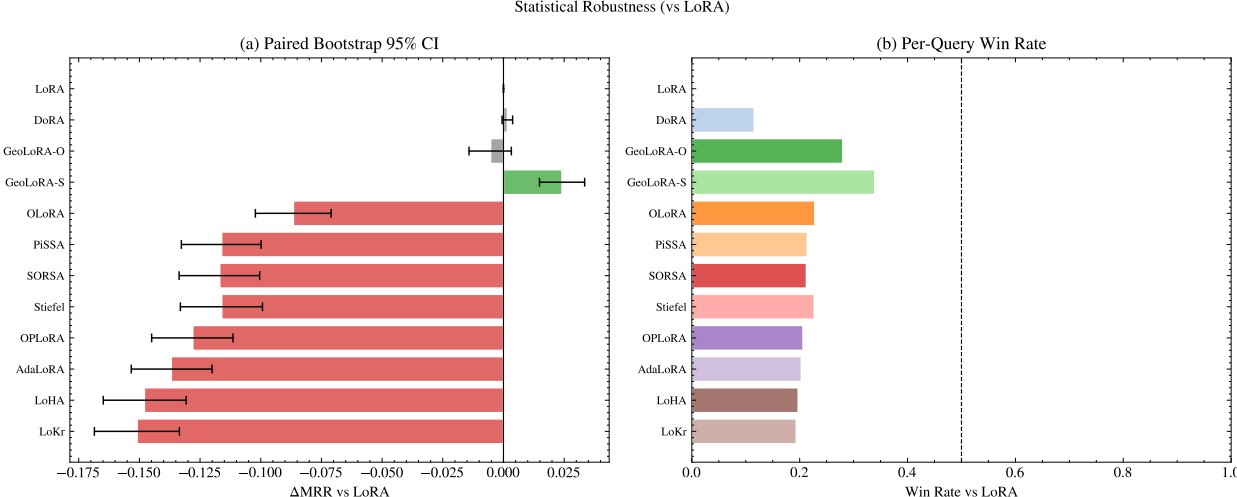

Figure 2: Per-query paired bootstrap comparison against LoRA on e5-small (3-seed, $N$=5,733 paired observations, 10,000 resamples). (a) 95% CI for $\Delta$MRR: GEOLoRA-SPECTRAL is the only method whose interval lies entirely above zero. (b) Per-query win rate: GEOLoRA-SPECTRAL wins on 31% of queries, highest among all methods.

**Across tasks.** Table 5 tests whether the ELSST pattern transfers to standard binary passage retrieval.[2] It does not transfer: on SciFact, DoRA leads; on NFCorpus, LoRA is best; and no orthogonal variant dominates both tasks. When relevance is binary and often signalled by lexical overlap, small changes in embedding geometry are less consequential. The spectral-alignment benefit therefore appears most relevant to tasks requiring fine-grained discrimination in a dense, multi-positive candidate pool—the properties that make ELSST geometry-sensitive.

Table 5: BEIR scope test on e5-small (MRR, 3-seed mean±std). The orthogonal advantage found on ELSST does not transfer to these binary passage-retrieval tasks.

| Method | SciFact | NFCorpus |
|---|---|---|
| *Unconstrained* | | |
| LoRA | 0.6925±0.0149 | **0.3400±0.0280** |
| DoRA | **0.7047±0.0056** | 0.3375±0.0299 |
| AdaLoRA | 0.6346±0.0050 | 0.1656±0.0006 |
| *Init-only orthogonality* | | |
| PiSSA | 0.6780±0.0092 | 0.3263±0.0077 |
| OLoRA | 0.6990±0.0135 | 0.2917±0.0153 |
| *Soft-constrained* | | |
| SORSA | 0.6780±0.0092 | 0.3263±0.0077 |
| *Hard-constrained* | | |
| Stiefel-LoRA | 0.6321±0.0238 | 0.3231±0.0571 |
| OPLoRA | 0.6645±0.0181 | 0.2780±0.0171 |
| GEOLoRA-ORTHO | 0.6501±0.0123 | 0.1964±0.0089 |
| GEOLoRA-SPECTRAL | 0.6747±0.0136 | 0.3113±0.0217 |
| *Structural counterfactual* | | |
| LoHA | 0.6375±0.0064 | 0.1685±0.0025 |
| LoKr | 0.6316±0.0009 | 0.1761±0.0044 |

---

[2]SORSA and PiSSA report identical BEIR scores because, under our shared recipe, the additional soft regulariser does not separate from the SVD-only initialiser on these tasks.

**Regime summary.** Combining the backbone and task evidence, our results suggest a narrower operating regime for GEOLORA-SPECTRAL: benefits are most plausible when $r/d$ is relatively large in our study, the task evaluates through cosine similarity in a semantically dense candidate pool, and the backbone is a compact encoder. Outside this regime, standard LoRA and DoRA remain strong default baselines.

## 6.2 Ablation Studies

**Rank sensitivity.** Table 6 varies $r \in \{2, 4, 8, 16, 32\}$ on e5-small. The $\Delta$MRR over LoRA is positive at every tested rank and is largest at $r=8$ (+0.0103), with a similarly strong gain at $r=2$ (+0.0097). Beyond $r=8$, the gap shrinks as rank increases, reaching +0.0027 at $r=32$. This is consistent with the idea that spectral preservation matters most when rank is a binding constraint: once the rank budget becomes large, even collapsed LoRA can retain enough capacity to perform well.

Table 6: Rank ablation on ELSST (e5-small, MRR, 3-seed mean±std).

| $r$ | LoRA | GeoLoRA-Spectral | $\Delta$ |
|----|------|------------------|----------|
| 2 | $0.4832 \pm 0.0033$ | $0.4929 \pm 0.0047$ | +0.0097 |
| 4 | $0.4987 \pm 0.0038$ | $0.5004 \pm 0.0063$ | +0.0017 |
| 8 | $0.5018 \pm 0.0065$ | $\mathbf{0.5121 \pm 0.0039}$ | +0.0103 |
| 16 | $0.5066 \pm 0.0018$ | $0.5128 \pm 0.0087$ | +0.0062 |
| 32 | $0.5075 \pm 0.0024$ | $0.5102 \pm 0.0046$ | +0.0027 |

**Initial sigma scale.** Among $\sigma_{\text{scale}} \in \{0.001, 0.01, 0.1\}$, 0.1 yields the best MRR in our tested range (0.5121); smaller values (0.4989 and 0.4957) make the spectral bridge too weak to exploit within the five-epoch budget.

**Computational overhead.** GEOLORA-SPECTRAL has the same leading-order parameter count as LoRA, namely $\mathcal{O}(r(d_{\text{in}} + d_{\text{out}}))$, plus an extra $r$-dimensional scaling vector per adapted projection. In other words, the additional parameters from $\boldsymbol{\sigma}$ are negligible relative to the directional factors $\mathbf{A}$ and $\mathbf{B}$. The main practical costs are a one-time truncated SVD at initialisation ($\sim 2\,\text{s/layer}$ in our implementation) and $\sim 5\%$ training wall-clock overhead from the orthogonal parametrisation. Inference cost after weight merging is identical to LoRA.

# 7 Discussion and Conclusion

## 7.1 Practical Guidance

Our experiments suggest the following practical heuristic. When adapting compact encoders for retrieval tasks whose quality depends strongly on embedding geometry, GEOLORA-SPECTRAL is worth including as a first-line baseline. When the backbone is larger or the task is closer to standard binary passage retrieval, the evidence in this paper does not support a consistent advantage for orthogonal variants; in those settings, LoRA and DoRA remain strong defaults.

## 7.2 Broader Implications

Three implications extend beyond the specific methods tested here.

First, the rank-collapse finding (Section 5.1) suggests that reported LoRA ranks in retrieval papers may overstate the effective capacity actually used by the adapter. Practitioners who select $r$ by grid search may therefore be searching over a narrower effective-rank range than the nominal values suggest. In our experiments, hard orthogonal constraints were the most reliable way to keep the update near its nominal rank budget.

Second, the failure of rank alone to predict retrieval quality (Section 5.2) cautions against using effective rank as a standalone diagnostic. The observed interaction with spectral alignment, coupled with the alignment paradox in which tighter positive pairs do not automatically yield better retrieval, suggests that embedding-space diagnostics should be multi-dimensional rather than reduced to a single scalar summary.

Third, the combination of orthogonality and spectral alignment (Table 3) connects to a broader theme in representation learning: the choice of *which* subspace to adapt can matter as much as the raw dimensionality of the update. This resonates with recent findings that the "primacy of magnitude" in low-rank adaptation depends on alignment with pretrained weight structures (Zhang et al., 2025b). Our retrieval-side evidence adds a geometric lens: when a task evaluates through cosine similarity in a dense candidate pool, misaligned updates can distort the global distribution even if they reduce the training loss.

### 7.3 Conclusion

Standard LoRA often collapses the effective rank of its updates, but restoring rank alone does not explain retrieval quality. Our evidence suggests that the strongest results on the compact, geometry-sensitive setting we study come from combining hard orthogonality with spectral alignment to the pretrained weight space.

More broadly, the paper argues that low-rank adapters for retrieval should be evaluated not only by end-task scores, but also by their geometric effect on the embedding space. The diagnostic toolkit used here, comprising effective-rank tracking, controlled 2×2 comparisons, and alignment–uniformity analysis, provides a starting point for analysing future embedding-centric adaptation methods.

### 7.4 Limitations

To contextualise our findings, we identify the following boundary conditions and limitations of our study:

**Task and Architectural Boundaries:** Our primary findings are anchored to ELSST, a deliberately geometry-sensitive benchmark. As shown by the BEIR scope tests, the advantages of hard orthogonality and spectral alignment do not universally transfer to binary passage-retrieval tasks where embedding geometry is less consequential. Likewise, the apparent dependence on $r/d$ is a hypothesis generated from three backbones rather than a calibrated threshold. Disentangling the relative effects of scale ($r/d$), architecture type (encoder vs. decoder), and lexical overlap remains open.

**Comparison Protocol:** To isolate parameterisation effects, we train all methods with a shared optimiser and schedule rather than reproducing each method's bespoke training recipe. The results therefore characterise behaviour under a controlled protocol, not the absolute best-case performance of every baseline.

**Hyperparameter Scope:** Our factorial analysis evaluates a fixed budget ($r \leq 32$) and a specific spectral bridge scale ($s = 0.1$). The behaviour of these methods at substantially higher ranks or at more extreme scaling factors remains unexplored.

**Mechanistic Completeness:** While the "alignment paradox" (GEOLORA-SPECTRAL retrieves better despite looser positive-pair alignment than GEOLORA-ORTHO) indicates that local tightness is a poor proxy for ranking quality, a complete mechanistic account of which geometric property ultimately drives the downstream gains is still emerging. This paper therefore offers a systematic empirical account rather than a formal proof that the identified orthogonal and spectral ingredients are uniquely optimal.

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
