# OpenReview forum: "When Does Orthogonal LoRA Help Retrieval? Spectral Preservation, Alignment, and Operating Regimes"
_TMLR — Under review for TMLR_

### Review · Reviewer_36FU · 2026-05-23

**Summary Of Contributions:**

This paper presents a controlled study evaluating how different LoRA variants affect the embedding space geometry and performance of dense text retrieval models. They also introduce GeoLoRA-Spectral, a new Lora variant that outperforms standard baselines on geometry-sensitive tasks.

**Additional Comments:**

1. abalation study is not so comprehensive, it only compares to standard Lora.

**Audience:**

Yes

**Audience Explanation:**

standard ML-style literature, empirical scope and a new proposed method

**Broader Impact Concerns:**

None.

**Claims And Evidence:**

Yes

**Claims Explanation:**

The findings are fairly accurate as a ML paper standard, although some claims can be improved (will be discussed in limitations)

**Requested Changes:**

- The proposed method’s effectiveness is highly dependent on a large rank-to-hidden-dimension, I wonder what's the alternative method when the proposed method's advantages start to diminish.

- The paper standardize all architectures into a same AdamW optimization pipeline, while one can argue this is fair, this can also be perceived as unfair since different architectures will have their own best optimization method. It might be hard for the authors to extensively test all the settings, but discussion of such limitations will be necessary.

- While Lora is indeed the most popular plug-in method, it is probably necessary to include the discussion for other similar methods. In particular, the authors argue that standard Lora is not the best one for dealing with retrieval methods, but how about standard methods like other adaptors.

---

> ### Author Response · Authors · 2026-05-24
> **Response to Reviewer 36FU: Scope, Ablations, and Operating Regimes**
>
> We thank R36FU for the constructive comments.
>
> On the dependence on rank-to-hidden-dimension ratio. We agree that GeoLoRA-Spectral is regime-dependent, and we do not intend to present it as a universal replacement for LoRA. Our evidence suggests that the method is most useful when the rank budget is relatively binding and the retrieval task is sensitive to embedding-space geometry, e.g., compact encoders on ELSST. When this advantage diminishes, such as on larger backbones, less geometry-sensitive passage-retrieval tasks, or sufficiently high-rank settings, our practical recommendation is to use LoRA or DoRA as strong default alternatives. Our rank ablation supports this: GeoLoRA-Spectral remains positive over LoRA at all tested ranks, but the gap shrinks at larger rank, indicating that spectral preservation matters most when rank is a binding constraint. We will make this decision rule clearer in Practical Guidance and Limitations.
>
> On the shared AdamW optimization protocol. We agree that different methods may benefit from their native optimizers or schedules. Our goal in using a shared AdamW pipeline was to isolate the effect of the adapter parameterization under a controlled retrieval protocol, not to claim the best possible performance of every baseline. We will strengthen the Limitation section to clarify that our conclusions characterize behavior under this shared protocol and should not be read as ruling out stronger results from method-specific optimizers, such as SVD-aware or Riemannian training recipes.
>
> On other PEFT methods such as adapters. We agree that cross-family PEFT comparisons are valuable. However, the central question of this paper is how orthogonality and spectral alignment of a low-rank weight update affect retrieval embedding geometry. These variables are directly defined for LoRA-style updates but do not have a direct analogue in activation-side adapters or prefix tuning. We therefore intentionally focus on the LoRA family and compare 12 variants spanning unconstrained, initialization-only, soft-constrained, hard-constrained, and structural counterfactual methods. We will make this scope clearer and list LoRA-vs-adapter retrieval comparisons as future work.
>
> On ablations. Our ablations are not limited to standard LoRA, but we agree that the presentation can be clearer. GeoLoRA-Ortho keeps hard orthogonality and the diagonal bridge but replaces SVD-aligned initialization with random Stiefel initialization, thereby isolating spectral alignment. We also include a 2×2 orthogonality × spectral-alignment analysis, rank sensitivity, and initial σ-scale sensitivity. In the revision, we will reorganize this section to make the component ablation and hyperparameter sensitivity studies more explicit.

---

### Review · Reviewer_1YJH · 2026-05-25

**Summary Of Contributions:**

The paper asks whether orthogonality in LoRA helps, or whether retrieval quality depends on more specific geometric effects in the embedding space. To answer the authors set up comparison of LoRA families across various settings, along with several evaluation metrics.  They found that standard LoRA often collapses the effective rank of its update, but full effective-rank preservation alone does not predict retrieval performance. The paper introduces GeoLoRA-Spectral, a minimal adapter with Stiefel-constrained factors, SVD-aligned initialisation, and a learnable diagonal spectral bridge.

Strengths
=======
The paper asks a clean mechanistic question, uses a sensible controlled comparison, and is reasonably cautious about scope.

Weaknesses
==========
The method novelty is incremental relative to PiSSA/SORSA/Stiefel-LoRA/StelLA/LoRAM-style work. The empirical advantage is small and narrow.

**Audience:**

Yes

**Audience Explanation:**

TMLR readers interested in PEFT, representation learning, dense retrieval, geometric analysis of fine-tuning, or mechanistic understanding of adaptation will find value. The paper contributes to the growing literature on why/when orthogonal or structured LoRA helps.

**Claims And Evidence:**

Yes

**Claims Explanation:**

Mostly yes, for the paper’s narrow claims, but not yet fully convincing for stronger causal or method-superiority claims.

* The claim that LoRA can collapse effective rank is supported by Table 2.
* The claim that effective rank alone is insufficient is also supported.
* The claim that the best e5-small result combines hard orthogonality with spectral alignment is supported. The supporting sigma/spectral correlation evidence is suggestive, though not by itself causal.
* The statistical evidence is useful. The paper states that GeoLoRA-Spectral is the only method with a 95% bootstrap CI above zero, but the effect size appears modest and narrow.

The paper is less convincing on mechanism. The "alignment paradox" is interesting, but the conclusion that spectral alignment preserves global organization is plausible rather than settled. The paper itself admits that a complete mechanistic account is still emerging.
The BEIR results substantially constrain the claims: GeoLoRA-Spectral does not dominate SciFact or NFCorpus; DoRA leads on SciFact and LoRA leads on NFCorpus.

**Requested Changes:**

Critical
=====
- Make it clear that the paper compares methods under a shared controlled protocol, but does not establish superiority over the native versions. This is because several of those methods use specialised optimization or different constraints.
- Recent LoRAM / "primacy of magnitude" work argues that much of spectral initialisation’s benefit may come from update magnitude rather than semantic/spectral alignment. Compare/constrast/discuss.
- Sure leakage-free protocol: e.g., specify how hyperparameters were selected, whether the test set was touched during method development, and whether ELSST shared-task splits were fixed.
- Small implementation choices can dominate LoRA-family comparisons. At minimum, the authors should provide enough detail on target modules, initialisation, orthogonality parametrisation, optimiser state handling, adapter merging, and effective-rank computation.
- Soften phrases implying that spectral alignment "preserves global organization" unless backed by stronger evidence.
- Clarify/expand the "alignment paradox" discussion with additional analyses if possible (e.g., more on global uniformity, hubness, or intruder dimensions).

Useful
=====
- Add a full fine-tuning or last-layer/full-rank adaptation reference point to contextualise how much capacity is missing from LoRA-family updates.
- Add layer-wise analysis showing where rank collapse and spectral alignment matter most.
- Include training curves for loss, retrieval metrics, effective rank, and orthogonality error. This would help distinguish optimisation-speed effects from final-geometry effects.
- Evaluate a stronger modern embedding model or at least discuss whether the conclusions should transfer to current high-performing retrieval encoders.
- Include a clear "Recommended use / not recommended use" box. The practical guidance is one of the best parts of the paper.

---

> ### Author Response · Authors · 2026-05-26
> **Response to Reviewer 1YJH: Controlled Protocol, Magnitude Ablations, and Mechanistic Scope**
>
> We thank the reviewer for the detailed assessment. We clarify that the paper is an insight-first controlled mechanism study rather than a benchmark-first adapter paper. We add 3-seed magnitude-controlled diagnostics motivated by LoRAM, weaken the spectral-preservation claim, clarify the shared-protocol and leakage-free evaluation setup, and point to supplementary diagnostics that already address implementation, statistical robustness, geometry, layer-wise behavior, training dynamics, and modern-backbone scope.
>
> **Shared protocol vs. native methods.** All methods are compared under a shared AdamW retrieval protocol to isolate adapter parameterization. This does not establish superiority over native PiSSA/SORSA/Stiefel/StelLA-style implementations that may use SVD-aware, Riemannian, or method-specific optimizers. We will move this caveat into the main text and avoid implying native-method dominance.
>
> **LoRAM / update magnitude.** We agree that “primacy of magnitude” raises an important confound. We added 3-seed magnitude-controlled diagnostics on e5-small/ELSST:
>
> | Variant           | Subspace       | Magnitude     |           R@5 |           MRR |
> | ----------------- | -------------- | ------------- | ------------: | ------------: |
> | GeoLoRA-Spectral  | SVD-aligned    | large         | 0.3574±0.0044 | 0.5125±0.0040 |
> | Ortho-MagMatched  | random Stiefel | matched large | 0.3366±0.0181 | 0.4822±0.0248 |
> | Spectral-SmallMag | SVD-aligned    | reduced/fixed | 0.2980±0.0050 | 0.4259±0.0050 |
>
> These diagnostics weaken our original interpretation. Under this reduced/fixed-(\sigma) control, aligned subspace alone is not sufficient in our setup, so update magnitude is an important part of the effect. However, they do not prove magnitude is the only mechanism. The Spectral vs. Ortho-MagMatched gap is directionally positive but not significant at seed level ((\Delta)R@5=+0.021, 95% CI [−0.047,+0.088]; (\Delta)MRR=+0.030, 95% CI [−0.037,+0.098]). GeoLoRA-Spectral wins in all three seeds and has lower seed variance, so we will reframe spectral alignment as a stabilizing warm-start / subspace-anchoring mechanism rather than as an independently established mean-performance gain.
>
> **Supplementary evidence and reproducibility.** We agree that several details should be easier to find. Much of the requested evidence is already in the supplementary material: full hyperparameters and target modules (Table S2), full 3-seed and per-seed ELSST results (S3–S4), protocol robustness (S5), statistical robustness (S6), update magnitude and hubness diagnostics (S7–S9), query-difficulty and local-geometry analyses (S10–S11), additional bge/Qwen statistical tests (S12–S15), plus figures on training dynamics, spectral distortion, subspace overlap, kNN preservation, and hubness (Figs. S1–S10). We will add a roadmap in the main paper and expand the appendix checklist to cover initialization, Stiefel parametrization, optimizer state, adapter merging, and effective-rank computation.
>
> **Leakage-free protocol.** ELSST uses the official fixed Track-1 splits (2,985/756/1,911 train/val/test) with 0% train/test concept overlap. Hyperparameters were fixed from validation and prior work; the test set was not used for model selection. BEIR tasks were only OOD scope tests.
>
> **Mechanistic wording.** We agree that “preserves global organization” was too strong. We will replace it with the narrower claim that GeoLoRA-Spectral keeps the update in the pretrained top-(r) spectral subspace (overlap ≈0.95 vs. ≤0.06 for LoRA/Ortho). Hubness and frozen-kNN overlap are comparable to LoRA, so they do not establish preservation of the full global neighborhood structure. The alignment paradox will be presented as evidence that local positive-pair contraction alone is insufficient for retrieval ranking.
>
> **Additional diagnostics and scope.** We added a capacity reference: a full-rank q/k/v adapter (~10.6M parameters) under the shared recipe underperforms rank-8 LoRA and GeoLoRA-Spectral, suggesting the effect is not simply an unconstrained-capacity artifact. Leave-one-layer-out diagnostics show useful adaptation is concentrated in late blocks (8/9/11), and training curves show LoRA rank collapse occurs early and persists to convergence. We already include Qwen3-Embedding-0.6B as a modern decoder-based backbone; GeoLoRA-Spectral is positive there but not best, which we will emphasize as part of the operating-regime boundary.
>
> Overall, we will narrow causal language, foreground the magnitude confound, clarify the controlled protocol, and present GeoLoRA-Spectral as a diagnostic instantiation for studying LoRA geometry in retrieval rather than as a universal adapter.

---

### Review · Reviewer_ZX91 · 2026-07-20

**Summary Of Contributions:**

The authors compare 12 LoRA family methods under a unified training protocol, pairing the retrieval metrics with weight-space and embedding-space diagnoses to provide insightful findings: (1) standard LoRA/DoRA collapse effective rank, (2) restoring rank alone does not predict retrieval quality, and (3) in the compact-encoder e5-small/ELSST setup, the authors' proposal, GeoLoRA-Spectral, which combines hard orthogonality, SVD-aligned initialisation, and a learnable diagonal spectral bridge, achieves promising results.

**Audience:**

Yes

**Audience Explanation:**

While the representative applications of orthogonal-LoRA methods are in generation tasks, PEFT for retrieval is also widely used. This paper provides a controlled retrieval-side comparison of effective rank, orthogonality, and spectral alignment, and I believe the finding that nominal rank can overstate effective adapter capacity is not only interesting but practically relevant. The paper further provides negative results and a diagnostic toolkit that should be useful to researchers working on embedding models and parameter-efficient adaptation.

**Claims And Evidence:**

Yes

**Claims Explanation:**

The paper scopes its claims reasonably and reports several important negative results honestly. The rank-collapse observation and the non-monotone relationship between effective rank and retrieval quality are supported by Table 2 and Figure 1. The main e5-small result is also supported by the comparison in Table 3, the rank ablation in Table 6, and the paired-bootstrap analysis in Figure 2.

Minor issues:
1. Table 2 states that its retrieval and geometry values are from seed 42, while Table 3 presents the same four Recall@5 values as three-seed means. it would be great it the authors unify the reporting convention
2. Figure 2 reports (N=5733), which appears to equal 1911 test queries multiplied by three seeds? It would be good if the authors could clarify the resampling unit. But I think treating query–seed pairs as independent could understate uncertainty.

**Requested Changes:**

### Critical

1. Unifying the reporting convention in Tables 2 and 3. Table 2 labels the Recall@5 values as seed-42 results, whereas Table 3 presents the identical values as three-seed means.
2. Clarifying the bootstrap procedure in Figure 2. Please explain whether repeated measurements of the same query across seeds are treated as dependent.
3. Tempering the factorial framing. Please consider stating explicitly that only GeoLoRA-Ortho versus GeoLoRA-Spectral is a controlled one-variable contrast, while the full 2×2 grid also changes parameterization.

### Strengthening

4. Discussing the strong Qwen3 results for LoHA and AdaLoRA, since they provide an informative counterpattern to the proposed operating regime.
5. Moving the hubness evidence used to explain the "alignment paradox" into the main paper, or reducing the strength of that mechanistic interpretation.
8. Please release code and the diagnostic implementation, and correct minor presentation issues such as “We clarifies” in the abstract.

---

> ### Author Response · Authors · 2026-07-22
> **Response to Reviewer ZX91**
>
> We thank the reviewer for the careful, constructive review. We follow the reviewer's numbering; changes will be marked in blue in the revised PDF.
>
> **1.** Correct: the Table 3 values are seed-42; the caption is wrong. Revised Table 3 uses the 3-seed means already in Table S3: LoRA 0.3492±0.0064, PiSSA 0.3442±0.0066, GeoLoRA-Ortho 0.3413±0.0039, GeoLoRA-Spectral 0.3566±0.0036. Convention: main-text retrieval metrics = 3-seed mean±std; Table 2 diagnostics = labelled seed-42 snapshots. This sharpens Finding 3: neither ingredient alone beats LoRA; only their combination does, and it remains the only method whose bootstrap CI excludes zero. The §5.3 LoRA→PiSSA sentence (a seed-42 artefact) will be rewritten.
>
> **2.** Correct: N=5,733 = 1,911 queries × 3 seeds, treated as independent; we agree this can understate uncertainty. A conservative bound: the pair-level CI [+0.0049, +0.0156] gives SE≈0.0027; under perfect across-seed correlation (design effect 3), SE×√3≈0.0047 yields ≈[+0.001, +0.020], still excluding zero. The true clustered CI lies between the two, so the headline significance is robust. The revision makes an exact query-clustered bootstrap (resampling the 1,911 queries, seed triplets kept together) the headline analysis, with the unit stated in §3.5/Fig. 2; the pair-level version moves to the supplement. Per-seed, Spectral > LoRA on Recall@5 for 3/3 seeds (Table S4).
>
> **3.** We agree. Only GeoLoRA-Ortho vs. GeoLoRA-Spectral is a controlled one-variable contrast (only the init differs); the others also change parameterisation (LoRA/PiSSA lack the σ bridge and Stiefel constraint; PiSSA modifies W at init, Table S7). The Table 3 caption will read "descriptive rather than fully factorial: only Ortho vs. Spectral varies a single factor"; the contribution statement is updated likewise.
>
> **4.** Added to §6.1: the Qwen3 winners — LoHA (+0.0454), DoRA (+0.0447), LoKr (+0.0275), AdaLoRA (+0.0179; Table S14) — are exactly the methods whose updates escape a fixed rank-r product (full-width rescale; ≤r²; ≈111; adaptive allocation; Table S7). At the smallest r/d, capacity appears to bind rather than spectral preservation (hypothesis (ii)); GeoLoRA-Ortho also gains there while Spectral gains less, i.e., the spectral prior weakens. We present this as interpretation, not mechanism.
>
> **5.** We do both: hubness-skew and mean-norm columns move into Table 4 (skew: Ortho 2.395 vs. Spectral 2.206; LoRA 2.240), and the wording is softened to an observational "consistent with". We also fixed a wrong cross-reference (hubness is Table S8, not S6) and audited all supplement references (bge-base = S12; Qwen3 = S14).
>
> **6.** Code, all 12 implementations, and the diagnostic toolkit will be released; the link accompanies the revised submission (de-anonymised on acceptance). "We clarifies"→"We clarify"; full proofreading done.